# A joint role for forced and internally-driven variability in the decadal modulation of global warming

Giovanni Liguori [1✉], Shayne McGregor [1], Julie M. Arblaster [1,2], Martin S. Singh[1] & Gerald A. Meehl [2]

Despite the observed monotonic increase in greenhouse-gas concentrations, global mean temperature displays important decadal fluctuations typically attributed to both external forcing and internal variability. Here, we provide a robust quantification of the relative contributions of anthropogenic, natural, and internally-driven decadal variability of global mean sea surface temperature (GMSST) by using a unique dataset consisting of 30-member large initial-condition ensembles with five Earth System Models (ESM-LE). We present evidence that a large fraction (~29–53%) of the simulated decadal-scale variance in individual time-series of GMSST over 1950–2010 is externally forced and largely linked to the representation of volcanic aerosols. Comparison with the future (2010–2070) period suggests that external forcing provides a source of additional decadal-scale variability in the historical period. Given the unpredictable nature of future volcanic aerosol forcing, it is suggested that a large portion of decadal GMSST variability might not be predictable.

[1] ARC Centre of Excellence for Climate Extremes, School of Earth, Atmosphere and Environment, Monash University, Melbourne, VIC, Australia. [2] National Center for Atmospheric Research, Boulder, CO, USA. ✉email: giovanni.liguori@monash.edu

I t is widely accepted that the observed long-term warming trend in global mean surface temperature (GMST) is primarily driven by increasing atmospheric greenhouse-gas concentrations[1]. However, our understanding of the decadal-scale fluctuations that are superimposed on this trend is still in its infancy. Many studies have highlighted the contribution of the internally generated Interdecadal Pacific Oscillation (IPO) and associated wind and sea surface temperature (SST) changes in the tropical Pacific as a major driver of GMST variability in both observations and models[2–8]. A recent study[9] suggests that internally generated variability, largely associated with the IPO, has played a significant role in decadal-scale fluctuations of GMST since the early twentieth century. Others argue that there could be a more limited role for unforced internal variability in twentieth-century multi-decadal (>30 years) climate variability, suggesting that such variability is primarily controlled by external forcing[10,11], or specifically that anthropogenic aerosols made a large contribution to the negative phase of the IPO, and thus the slowdown in GMST trend in the early 2000s[12]. However, the relative contributions of natural variability and different external forcing agents, such as anthropogenic and volcanic aerosols, continue to be debated[8].

Given the chaotic nature of the climate system and the single trajectory we observe, climate model simulations, such as the Coupled Model Intercomparison Project phase 5 (CMIP5), have been often used to separate forced and internally generated decadal-scale variability[13]. Multi-model ensembles have primarily been used for this purpose, but this approach is limited because of the apparent model-dependent response to external forcing[14,15]. Averaging across single (or limited numbers of) ensemble members of multiple models acts to conflate the anthropogenic and internally driven variability of each model. While forced variability has been isolated in single-model ensembles[16], no prior study has presented robust estimation of forced variability using large ensembles with multiple models.

In the present study, we leverage large initial-condition ensembles from five models to offer evidence for a prominent role of external forcing in modulating decadal-scale (i.e., time-scales between 8 and 32 years) fluctuations of GMSST that are superimposed on the accelerating warming trend. While the fraction of externally forced variance in GMSST varies among models, all simulations present synchronised decadal fluctuations of GMSST that are largely driven by volcanic eruptions. Our results also suggest that the IPO, which is largely thought to be internally generated, potentially contains a forced component, implying that past phase transitions of the Pacific climate could be a predictable response to external forcing.

## Results

**Earth System Model large ensembles.** This study uses simulations from the US Climate Variability and Predictability programme's newly developed data archive of large initial-condition ensembles with Earth System Models (hereafter ESM-LE[17]). All ESM-LE simulations utilised similar CMIP5 forcings, which include anthropogenic and volcanic aerosols, solar radiative forcing, and greenhouse-gas (GHG) concentrations that represent historical radiative forcing up to 2005 and the Representative Concentration Pathway 8.5 thereafter[18]. For this study, we use only models that provide at least 30 members, namely CANESM2, CESM1, CSIRO, MPI, and GFDL-ESM2M (hereafter simply GFDL; see Supplementary Table 1 for details about the ensembles). The large number of members of each single-model ensemble allows us to effectively isolate the forced component from the internally generated component of variability in each model, while the comparison between

historical (1950–2010) and future (2010–2070) periods allows us to investigate fundamental differences in the forcing used in the two periods (e.g., the absence of volcanic forcing in the latter period).

**Forced GMSST decadal variability.** To place our findings in the context of the IPO, we analyse SST rather than surface air temperature. While results using either field are almost identical, this choice facilitates the comparison with Pacific climate modes that are typically calculated from SST. To isolate the forced component of the decadal-scale variability of global mean SST (GMSST) in each ESM-LE model, we compute ensemble means of 8-year low-passed time series of area-weighted GMSST by averaging across each single-model ensemble (Fig. 1a). Due to the lack of long-term reliable observations in polar regions and before 1950, we limit our analysis spatially to the near-global domain, here defined as regions between 40°S and 60°N[19], and temporally from 1950 onward. While the amount of warming reached toward the end of the twenty-first century varies among models (Fig. 1a), the GHG-induced long-term trend in each model is well approximated by a quadratic function, especially after the year 2000 (Supplementary Fig. 1). Since we are interested in decadal-scale fluctuations of GMSST, we subtract the quadratic fit to the ensemble mean from each GMSST ensemble member to obtain residual time series (GMSSTr). These residual time series highlight variability largely independent of centennial length anthropogenic change (Fig. 1b). Moreover, the removal of a centennial-scale quadratic trend does not significantly affect the shorter decadal-scale fluctuations targeted in this study, apart from possible effects at the edges of the study period (i.e., 1950 and 2070).

Two clearly distinct periods are identified in Fig. 1b: from 1950 to 2010, when models show decadal-scale fluctuations that are largely coherent with each other, and from 2010 to 2070, when models show incoherent variability with model spread more closely centred around zero (coloured shading in Fig. 1a–c). The observed GMSSTr trajectory lies largely within the ESM-LE model spread envelopes and presents some similarities with GMSSTr ensemble means from about 1975 to 1995, with the lack of full agreement largely due to the internal variability, but perhaps also due to model deficiencies in representing the external forcing[20] (discussed later). Given the influence of GHG forcing has largely been removed via the quadratic fit, these results suggest that non-GHG forcing is responsible for synchronising decadal-scale variability in ESM-LE ensemble means, and potentially the observations, over the historical period (1950–2010). Despite the consistency, there are important differences between the models and observations. Neglecting any residual internal variability in the ensemble mean, the correlation between the ensemble mean and each individual member gives a direct measure of the GMSSTr forced component for each model (Supplementary Fig. 2). Depending on the model, the forced component explains between 30 and 58% of the time-evolving variance (explained variance obtained as correlation squared) in the historical period and 8 to 18% in the future period, with the remaining variance associated with unforced internal variability. However, three out of five models show correlations between observed and ensemble-mean time series outside the range of modelled internal variability (i.e., outside the range of correlations between the ensemble mean and individual ensemble members). This may be a result of models underestimating the range of natural variability, but, as we show below, it is likely to also indicate an overestimation of the forced component in the historical period (e.g., Supplementary Fig. 3).

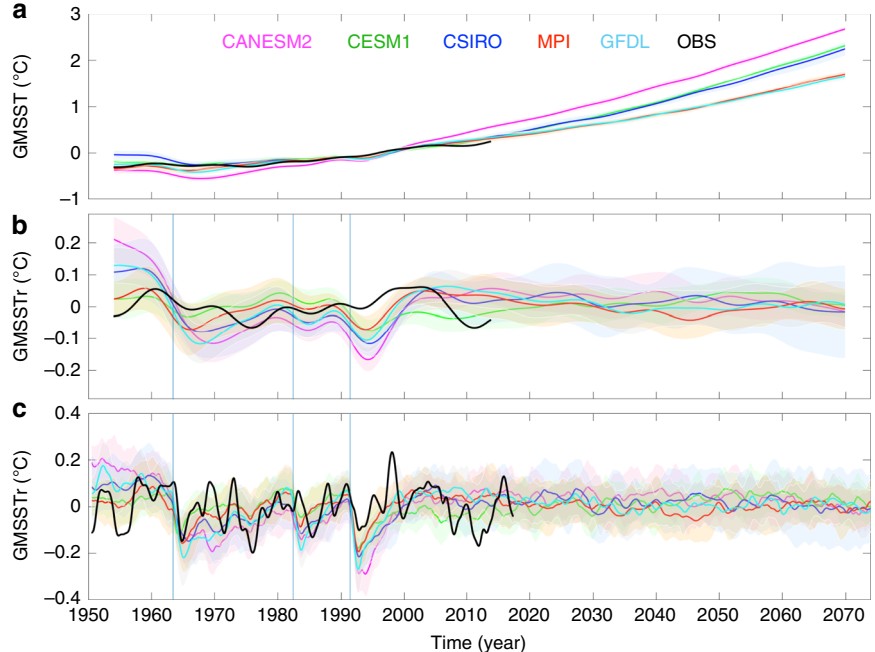

**Fig. 1 Decadal variability in global mean sea surface temperature. a** Global mean sea surface temperature (SST) anomaly (GMSST) time series computed from 8-year low-passed SST for CANESM2 (magenta), CESM1 (green), CSIRO (blue), MPI (red), GFDL (cyan), and observations (OBS; black). **b** Decadal-scale fluctuations in GMSST as highlighted by the GMSST residual (GMSSTr) obtained removing the quadratic fit from the time series shown in **a** (see Supplementary Fig. S1 for details on the quadratic fit). **c** Same as **b** but using 1-year low-passed SST. GMSST is defined as area-weighted mean SST between 40°S and 60°N and anomalies are relative to the 1980–2010 mean. Colour shading indicates the one standard deviation model spread envelope and units are in (°C).

In addition, we estimate the fraction of forced decadal-scale variance (FDV) in GMSSTr by comparing the power spectrum of the ensemble-mean GMSSTr time series, which represents the externally forced variance at each frequency (thick coloured curves in Fig. 2), with the average power spectrum of each ensemble member's GMSSTr time series, which represents the sum of internally and externally forced variance (i.e., total variance) at each frequency (thick grey curves in Fig. 2). Specifically, integrating both power spectra for periods between 8 and 30 years (i.e., decadal-scale variance) reveals that the FDV in each ESM-LE accounts for 29–53% of the total GMSSTr variance of the historical period (Fig. 2a, c, e, g, i). In contrast to the historical period, the FDV in the future period is greatly reduced in all models as shown by the nearly flat power spectrum of the ensemble mean GMSSTr (Fig. 2b, d, f, h, l) and the smaller fraction (i.e., 3–6%) of decadal variance in GMSSTr accounted for by FDV for the years 2010–2070.

Furthermore, larger oscillations in ensemble mean and model envelopes (solid lines and colour shading in Fig. 1b, respectively) in the historical period compared to the future period suggest that the total decadal-scale variance is also larger in the former period. In fact, the power in the GMSSTr ensemble-mean time series between 8- and 30-year periodicity over the historical period (1950–2010) is between 1.8–4.5 times larger than in the future period (2010–2070). This indicates that external forcing provides an additional source of decadal-scale GMSST variance in the historical period. However, it must be noted that the observed power spectrum (black lines in Fig. 2) for periods between 8 and 30 years is lower than almost all ensemble members (grey lines in Fig. 2), independently of the model. While it is possible that the observed GMSSTr trajectory is a rare event even with these multiple large ensembles, the discrepancy is more likely a symptom of the model tendency to overestimate internal and/

or non-GHG forced decadal variability (e.g., Supplementary Figs. 2 and 3).

**Role of aerosols, GHG, and biomass burning in decadal variability.** A possible cause of the distinct character of past and future decadal-scale variability in ESM-LE appears evident when the width of the GMSSTr low-pass filter is reduced to 1 year (Fig. 1c). Year-to-year variations in GMSSTr reveals three abrupt drops in temperature that coincide with major volcanic eruptions: Agung in March 1963, El Chichón in April 1982, and Pinatubo in June 1991 (vertical lines in Fig. 1b, c). While it is well known that stratospheric aerosols associated with explosive volcanic eruptions can have short-term cooling effects on global mean temperatures in observations[21,22] and models[23], and likely contributed to the decadal-timescale slowdown in the rate of global mean temperature warming in the early 2000s[24], results here demonstrate their important role as drivers of historical decadal-scale GMSST variability in current state-of-the-art climate models (i.e., ESM-LE). The temporal spacing between these strong drops in temperature, 19 and 9 years, combined with a recovery time (i.e., time to dissipate the cold anomaly) of 5–8 years[25] (Fig. 1c), creates a climate signal with a strong projection on decadal-scale variability.

The key role of volcanic eruptions for FDV is indirectly confirmed from CESM1 single-fixed-forcing experiments[26] in which concentrations of either GHGs, tropospheric inorganic aerosols, or organic aerosols associated with biomass burning are fixed to their 1920 values. All other forcing components, including volcanic forcing, use time-dependent values as in the full-forcing CESM1 simulations of ESM-LE. Comparing full-forcing with single-fixed-forcing ensembles reveals only minor changes in the ensemble mean of 8-year low-passed GMSSTr (Fig. 3), with correlation coefficients as high as 0.86, 0.88, and

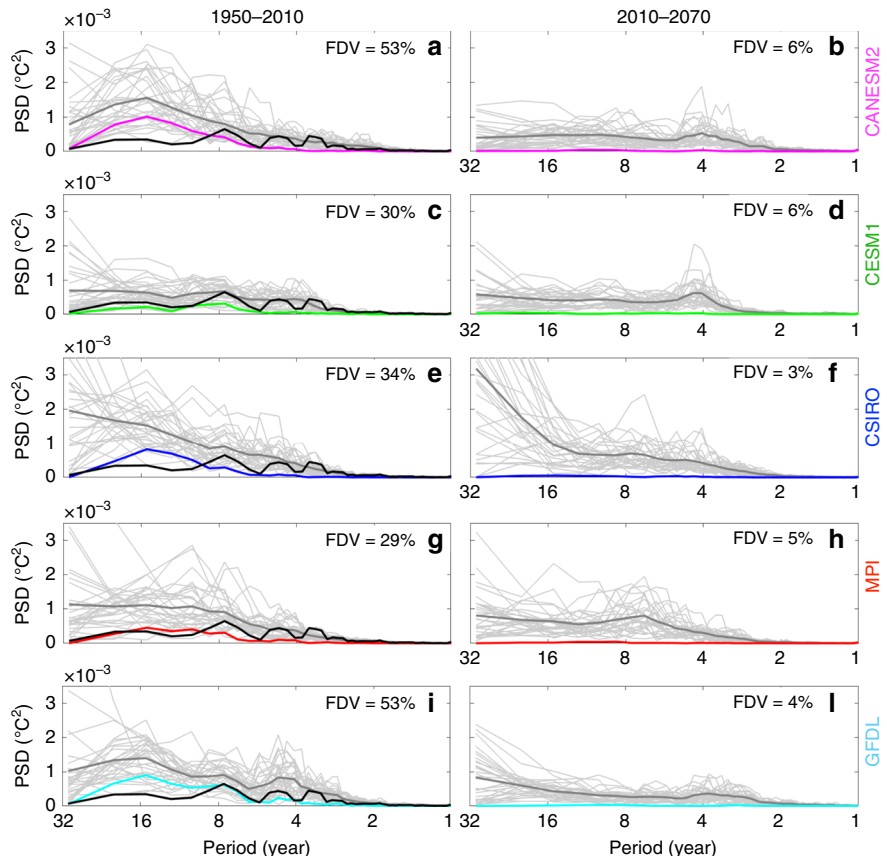

**Fig. 2 Forced versus internally generated decadal variability.** Power spectral density (PSD) of global mean sea surface temperature residual (i.e., with quadratic fit removed) for past (1950–2010) and future (2010–2070) decades for models in the ESM-LE as given by the coloured labels. Each panel shows the power spectrum for the ensemble mean (thick coloured line) and for each ensemble member (thin grey line). Thick grey line shows the average of the power spectrum across the ensemble (i.e., the average of the thin grey lines), and thick black line on the left panels shows the observed power spectrum. The number in each subfigure indicates the percentage of forced decadal-scale variance (FDV; powers integrated between 8- and 30-year periods) computed as the ratio between the ensemble mean power spectrum (i.e., forced variability; thick coloured) and the average power spectrum (i.e., total variability; thick dark grey). Units are in ($°C^2$).

0.62, for fixed GHGs, fixed tropospheric inorganic aerosols, and fixed biomass-burning aerosols, respectively.

All simulations include time-dependent volcanic forcing, and all simulations exhibit coherent decadal-scale variability of GMSST.

A direct estimate of volcanic-driven GMSST variability requires large ensembles of simulations in which the volcanic forcing is either the only one present (i.e., volcanic-forcing-only experiment) or the only one excluded (i.e., all-but-volcanic-forcing experiment). While such experiments are unavailable in CESM-LE, results from smaller ensembles (four and five members) with a similar version of CESM are consistent with the hypothesised role for volcanic forcing (Supplementary Fig. 4).

While fixing biomass-burning aerosols seems to have the largest impact on GMSSTr trajectories (least correlation with the full-forcing experiment), it must be noted that the fixed biomass-burning ensemble has only 15 members, and thus the ensemble mean retains more random internal variability than in the other single-fixed-forcing ensembles that have 20 members. It is noteworthy that fixing tropospheric anthropogenic aerosol concentrations to their 1920 value results in larger amplitude fluctuations in GMSSTr, suggesting that higher tropospheric aerosol concentrations typical of the second half of the twentieth century have a damping effect on the volcanic forcing. While this needs further confirmation in other models, it suggests that anomalies in stratospheric aerosol concentrations associated with

volcanic eruptions have a larger impact in a cleaner atmosphere; determining the reasons for this is left for future work.

Comparing GMSST anomalies after each major volcanic eruption suggests a tendency for a larger-than-observed response in ESM-LE models during the 1982 and 1991 events (Supplementary Fig. 3). A similar discrepancy in CMIP5 simulations has been reconciled by taking into consideration the concomitance of these eruptions with El Niño events; the positive anomaly in GMSST associated with the El Niño partially offsets the cooling associated with volcanic eruptions in observations but not in simulations[27]. Here with 30 members for each model we see that in most ESM-LE simulations, and in all members from CANESM2 and GFDL ensembles, the magnitude of the global temperature response to the 1982 and 1991 volcanic events is overestimated regardless of the El Niño-Southern Oscillation phase (Supplementary Fig. 3). Furthermore, models that likely overestimate the volcanic response, CANESM2 and GFDL, present also the highest percentage of decadal variability accounted for by FDV (53%; Fig. 2a, l), suggesting that these models might overestimate the externally forced fraction of decadal variability, which is therefore closer to the lower boundary of the range estimated in ESM-LE (i.e., 29–53%).

**Forced variability of the IPO.** While we have focused on the temporal variability in GMSSTr, the spatial expression of

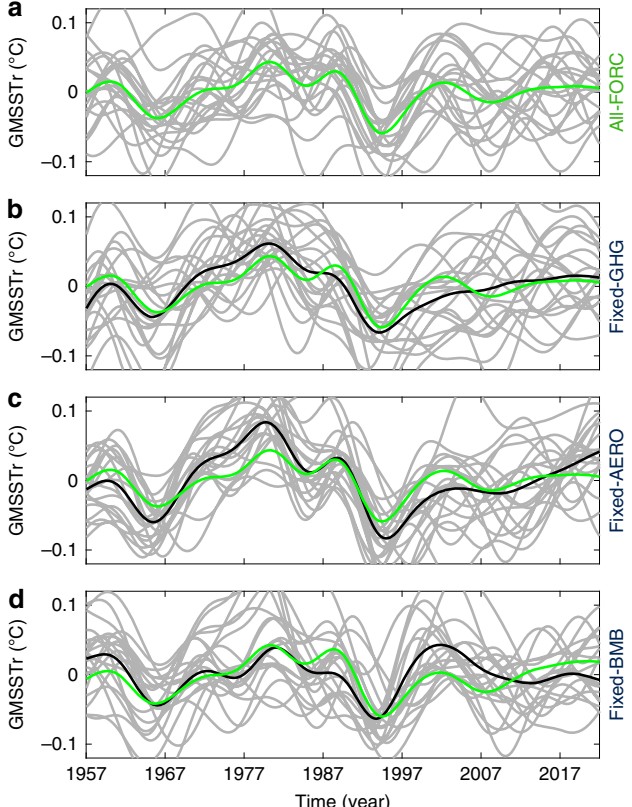

**Fig. 3 Role of the external forcing in CESM1.** Global mean sea surface temperature anomaly residual (GMSSTr) time series for each ensemble member (grey) and for the ensemble mean (black) in single-fixed-forcing ensembles. **a** All forcing (All-FORC, 20 members). **b** All forcing but fixed greenhouse gases (Fixed-GHG, 20 members). **c** All forcing but fixed anthropogenic aerosols (Fixed-AERO, 20 members). **d** All forcing but fixed biomass burning (Fixed-BMB, 15 members). In each panel, green line shows the ensemble mean in the All-FORC case. Units are in (°C).

GMSSTr during 1950–2010 uncovers an important characteristic of the FDV in both ESM-LE and observations. Specifically, the ensemble-averaged regression map of GMSSTr onto 8-year low-passed SST reveals a strong resemblance to the IPO pattern (Fig. 4a), which results in a spatial correlation with the IPO pattern (see Fig. 4 caption for IPO definition) of 0.42 in the observations and of 0.76, 0.80, 0.75, 0.87, and 0.52, in the ensemble mean of CANESM2, CESM1, CSIRO, MPI, and GFDL, respectively. Consistent with this spatial correlation, GMSSTr time series are also significantly correlated with the IPO index in several ESM-LE simulations (Supplementary Fig. 5). While the spatial and temporal correlation between GMSSTr and the IPO over the historical period may simply confirm the known contribution of IPO variability to twentieth century global surface temperature fluctuations[7], it may also indicate the presence of an externally forced component in the IPO variability resulting from a partial synchronisation of the internal variability.

The possibility of a forced component in the IPO variability[12,28] finds partial support in the variance associated with the IPO computed from ensemble mean SSTs as a function of the ensemble size (Fig. 4c). Assuming the IPO to be purely a result of internal variability, one would expect this IPO variance to decrease in proportion to the number of ensemble members, $n$ (see "Methods"). However, in CANESM2 and CSIRO, and partially in GFDL, the fraction of variance linked to the IPO during 1950–2010 decreases at a lower rate and retains a stable

residual variance (indicated in Fig. 4c as RV) of 15%, 13% and 7%, respectively, of the initial value (i.e., variance for $n = 1$) for $n > \sim 20$. While this sizeable residual in CANESM2 and CSIRO is a clear indication of a forced component of the IPO during the historical period, repeating the exercise for the years 2010–2070 shows a reduction of variance that approximates well the expected $1/n$ decrease in all five ensembles, suggesting that virtually all the IPO variability in the future period is internally generated.

## Discussion

Using multiple large ensembles, we present evidence of an externally forced component in the decadal variability of GMSST that is largely driven by volcanic eruptions and unrelated to GHG forcing. This result is consistent with a recent study by Haustein et al.[10] in which virtually all of the multi-decadal variability in GMST was reconstructed by combining forcing time series with a simple impulse response model. However, our different methodological approach provides new and independent estimations for the forced component that is free from potential overfitting issues that may affect the conclusions in Haustein et al.[10]. While the forced variability is visible in the observations, it must be noted that models appear to show too much decadal variability overall (Fig. 2) and likely overestimate the response to the non-GHG external forcing (in particular volcanic aerosols) on decadal timescales (Supplementary Figs. 2 and 3). In addition to GMSSTr, our findings suggest that the non-GHG external forcing might also modulate variability in the IPO. The view of a forced IPO component is consistent with previous studies[12,28], but we further show that the forced pattern in the Pacific (see GMSSTr regression maps in Fig. 4b) has a strong resemblance to the IPO pattern by separating externally and internally driven components.

While we have focused on the IPO, which is one expression of Pacific decadal variability (PDV)[29], it must be noted that external forcing might play an even more important role in SST variations in the Atlantic, where spatiotemporal variability in anthropogenic aerosol emissions may be driving most of the simulated Atlantic multi-decadal variability (AMV)[30,31]. The possible existence of forced components in PDV and AMV provides an additional explanation to our main finding that historical fluctuations in simulated decadal-scale GMSST variability have a large forced component (i.e., 29–53% of explained variance). Given the unpredictable nature of future volcanic eruptions, these results confirm the significant caveat that must accompany decadal climate predictions that would be significantly affected by a volcanic eruption[32,33].

## Methods

**Significance levels of the correlation coefficients.** Unless otherwise stated, the significance of the correlation coefficients throughout the study is estimated by computing empirical probability density functions (EPDFs) for the correlation coefficient of two red-noise time series, which have the same lag-1 autoregressive coefficient of those estimated in the original signals. We assess the 99% significance levels using an EPDF obtained from 10,000 realisations of random red-noise time series.

**A null hypothesis for the IPO variance in the ensemble mean.** Assuming the IPO to be purely a result of internal variability, thus independent in each ensemble member, one can use a known statistical result to predict the ensemble mean IPO variance as function of the number of ensemble members.

Consider $N$ independent time series denoted $X_i$, $i = 1, 2, \ldots N$, each with variance $\text{Var}(X_i) = \sigma^2$, $i = 1, 2, \ldots, N$. The variance for the average of the $N$ time series is $\text{Var}(\frac{1}{N}\sum_{i=1}^{N} X_i) = \frac{1}{N^2}\{\sum_{i=1}^{N}\text{Var}(X_i) + \sum_{i \neq j}^{N}\text{Cov}(X_i, X_j)\}$, where $\text{Cov}(X_i, X_j)$ is the covariance between the time series $X_i$ and $X_j$, $i, j = 1, 2, \ldots, N$. Since the time series are assumed to be independent, this last term equal to zero and the variance for the average reduces to $\text{Var}(\frac{1}{N}\sum_{i=1}^{N} X_i) = \frac{1}{N^2}N\sigma^2 = \frac{\sigma^2}{N}$, namely, the variance for each independent time series, $\sigma^2$, scaled by $N$.

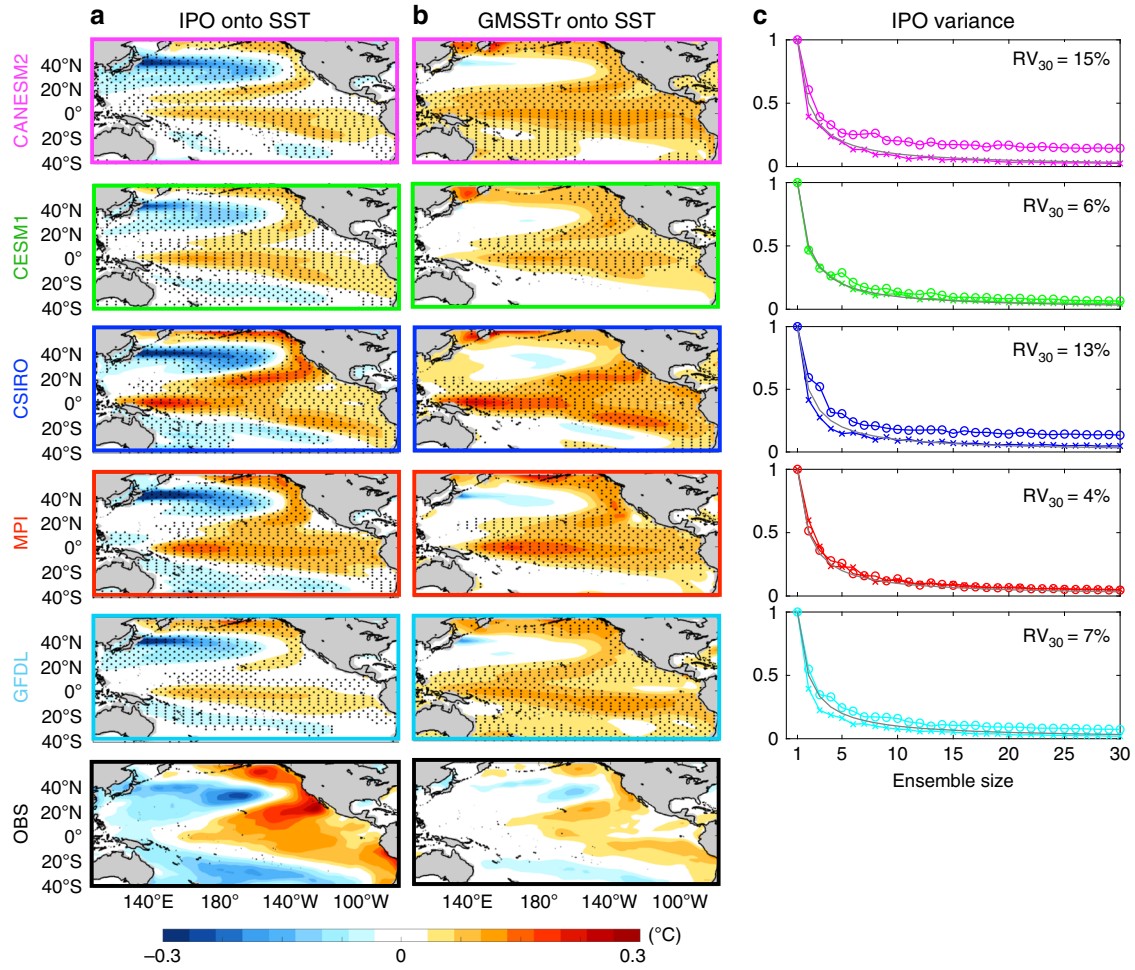

**Fig. 4 Forced decadal fluctuations and Interdecadal Pacific Oscillation (IPO) variability. a** Regression maps of IPO index onto 8-year low-passed sea surface temperature (SST) during past decades (1950–2010) for the ESM-LE ensemble mean and observations. **b** Same as **a** but for global mean SST residual (GMSSTr) time series. **c** Ratio of the variance of the ensemble mean IPO index to the mean variance of the IPO index in individual ensemble members as a function of the ensemble size $n$ during past (1950–2010; coloured circles) and future (2010–2070; coloured crosses) periods. Dark grey lines in each subfigure in **c** show the expected $1/n$ decay for the variance of the sum of $n$ independent random variables with equal variance. Each subfigure also indicates the residual variance (RV) for $n = 30$ in the historical period. IPO index defined as 8-year low-passed Tripole Pacific Index (TPI) following Henley et al.[35]. The TPI is calculated as the difference between the SST anomaly averaged over the central equatorial Pacific (10°S–10°N, 170°E–90°W) and the average of the SST anomaly in the Northwest (25°N–45°N, 140°E–145°W) and Southwest (50°S–15°S, 150°E–160°W) Pacific. The stippling over the maps in **a**, **b** indicate anomalies larger than one standard deviation of the inter-member spread.

## Data availability

We use SST observation from National Oceanic and Atmospheric Administration Extended Reconstruction SST, version 3 (ERSST v3) product[34], which consists of monthly mean values from 1854 to the present on a 2° × 2° horizontal grid globally.

Details about ESM-LE model outputs collected from the US CLIVAR Working Group on Large Ensembles can be found at http://www.cesm.ucar.edu/projects/community-projects/MMLEA/. In addition to ESM-LE, we use three single-fixed-forcing CESM1 20-member ensembles produced using the same model configuration, grids, and inputs of the ESM-LE CESM1 ensemble, available from http://www.cesm.ucar.edu/working_groups/CVC/simulations/cesm1-single_forcing_le.html. The CESM1 volcanic-only ensemble was submitted to CMIP5 and is available via ESG from https://esgf-node.llnl.gov/search/cmip5/; the CESM1 all-but-volcano ensemble is available https://doi.org/10.26024/t1a4-tk97.

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

## Acknowledgements

We acknowledge support from the Australian Research Council through the Centre of Excellence for Climate Extremes (CE170100023). Computational resources and services from the National Computational Infrastructure (NCI), which is supported by the Australian Government, are gratefully acknowledged. We also thank the US CLIVAR Working Group on Large Ensembles for collecting and making available the multi-model large ensemble archive (ESM-LE, http://www.cesm.ucar.edu/projects/community-projects/MMLEA/). We thank the National Center for Atmospheric Research for producing and making available both the results from the CESM Large Ensemble Project and the single-fixed-forcing experiments (http://www.cesm.ucar.edu/projects/community-projects/). Portions of this study were supported by the Regional and Global Model Analysis (RGMA) component of the Earth and Environmental System Modelling Programme of the U.S. Department of Energy's Office of Biological & Environmental Research (BER) via National Science Foundation IA 1844590 and under Contract DE-AC52-07NA27344 to Lawrence Livermore National Laboratory. This work also was supported by the National Center for Atmospheric Research, which is a major facility sponsored by the National Science Foundation under Cooperative Agreement No. 1852977. This research used resources of the National Energy Research Scientific Computing Center, a DOE Office of Science User Facility supported by the Office of Science of the U.S. Department of Energy under Contract No. DE-AC02-05CH11231. G.L. also thanks F. Lehner and E. Di Lorenzo for insightful discussions in the early and late stages of this work, respectively, and J. Champlin for proofreading the manuscript at various points throughout its progress.

## Author contributions

G.L. conceived and designed the study, analysed the data and wrote the paper. S.M., J.M.A., M.S.S. and G.A.M. designed the study and wrote the paper. All authors contributed to the discussion of results.

## Competing interests

The authors declare no competing interests.
