## [Peer Review File · Nature Communications]

Reviewers' comments first round:

Reviewer #1 (Remarks to the Author):

Review of the manuscript "A joint role for forced and internally-driven variability in the decadal modulation of global warming" by G. Liguori et al.

This paper presents the effects of volcanic aerosol forcing in the decadal variations of global mean surface temperature. By using large initial-condition ensembles with Earth System Models, they separate anthropogenic, natural, and internally-driven components of decadal variability. They found the volcanic aerosols account for ~29-53% of the decadal variance over 1950-2010, while externally forced decadal variance is very weak during 2010-2070 due to the lack of volcanic forcing. In addition, they also confirm the possible projection of external forcing onto the IPO variability. The paper is well-written and well-organized. However, some drawbacks should be addressed before the paper can be considered for publication in Nature Communications. Thus, the paper needs a major revision.

Major comments:

1. In this paper, both IPO and PDV are used to describe the decadal variability in the Pacific. Do you regard them as the same thing in this paper or not? Why do you use PDV in some places but IPO in others? Please clarify. For example, in Lines 71-76, the two sentences provide similar information, but the former uses IPO while the latter uses PDV. What is the difference between IPO and PDV?
2. Besides the effect of volcanic forcing, the importance of internal variability can be inferred in the results of this paper. I suggest the authors adding some analysis and discussion to compare the contributions from internal component and volcanic forcing to the decadal modulation of GMSST quantitatively, not only for the variance as you have mentioned but also for the time evolution.
3. By comparing between historical and future periods, the effect of volcanic forcing is discussed. Besides volcanic aerosols, are there any other forcing differences between the two periods that can cause differences in decadal variability? How about the anthropogenic aerosol forcing? Are there any decadal variations in anthropogenic aerosol forcing in historical or future periods? Please clarify.
4. Lines 126-128: The decadal variations seem not very similar between observation and ensemble means. Why are the decadal variations similar among models, but different from the observation? They should have the same externally forced variations. Does this difference indicate the effect of internal component in the observed variability? The correlation coefficients between observation and ensemble means can indicate the percent of decadal variations explained by volcanic forcing.
5. Lines 162-165: Similar to the last comment, the decadal variations from volcanic eruptions are obvious only in the ensemble means but not in observation, again indicating the importance of internal variability in observation. This should be clarified in the text.
6. Lines 180-183: This statement is interesting and needs to be clarified. Why aerosols from troposphere have offset effect to those from stratosphere? Do you have any conjecture or related references to support this statement?
7. Lines 216-217: "Assuming the IPO to be purely a result of internal variability, one would expect this IPO variance to decrease in proportion to the number of ensemble members, n ." How did you get this expectation? I cannot understand. Please explain.
8. Fig. 3: Why don't you provide the results for simulations with fixed volcanic forcing here? That will be more straightforward to prove its effect in dominating the decadal variations in GMSST.

Minor comments:

- 1 Lines 216 and 220, Fig. S2 should be Fig. 4c.
- 2 Caption of Fig. 3: The second "b" should be "d"

Reviewer #2 (Remarks to the Author):

Review of "A joint role for forced and internally-driven variability in the decadal modulation of global warming" by Liguori et al.

This manuscript examines how much decadal-timescale variability in Earth system model Large

Ensembles is due to external forcing and how that amount compares to the decadal variability in observations. By using Large Ensembles in both the historical and future period, the authors are able to assess how much decadal-timescale variability is externally forced (using the ensemble mean) versus internally generated (by examining all ensemble members). They trace much of the decadal variability in modeled global warming during the historic period to volcanic aerosols. The addition of results from the CESM all-but-one-forcing ensembles further illustrate this possibility in a convincing way. In these Large Ensembles, volcanic eruptions tend to excite the IPO pattern, indicating that much variability in the IPO may be externally forced. From there, the manuscript concludes that modeled decadal variability in the future may be unpredictable given the unpredictable nature of volcanic eruptions.

I very much appreciate the approach of using multiple Large Ensembles. For studies examining climate variability, this approach is still quite novel (we've only had convenient access to multiple Large Ensembles for the last year) and should be a gold-standard method going forward for examining these kinds of questions in climate models. This method gets at both models' representation of internal variability and allows an assessment of structural model differences, as the authors have done here. Because of the rigor of using Large Ensembles along with strong statistical methods, I find the conclusions drawn from this study convincing and of relevance to the broader climate community. My one suggestion to the authors is to further flesh out another possible conclusion that is implied at multiple points in the manuscript: that climate models as a whole are reacting too strongly to aerosols and that forced IPO variability is too strong overall as compared to observations (see discussion below). In the figures, I see some support for this argument and think that the authors would only strengthen their manuscript by pointing out this possibility more explicitly.

The figures are of publication quality and the writing is very clear with almost no typos. I recommend acceptance after addressing a few (very) minor concerns and suggestions, listed below.

Minor concerns/suggestions:

1. As discussed above, I think that the results of this manuscript suggest that all of these ensembles have too strong responses to volcanic aerosol forcing (besides just CANESM2 and GFDL) and that there is a structural model deficiency across all the ensembles in the same direction. I suggest to the authors to address this possibility a bit more in the conclusions section. Here's the evidence I see in the figures for this possibility:
 - Fig 1b/c, 1991. The observed dip in residual GMSST is at the very upper edge or outside the spread of all ensembles. While it is possible that 1991 was a rare event, the fact that observations lie outside the spread of all 5 ensembles (in 1b) suggests to me that all 5 ensembles have something in common that is leading them to have a similar (possibly incorrect) response.
 - Fig 2, left column: The observed power spectral density (black lines) is lower than the ensemble mean (forced) variability and lower than almost all the ensemble members (gray lines) in all 5 ensembles for periods ≥ 8 years. I find it rather unlikely that across all 5 ensembles with 150+ possible realizations, that the observed realization at 8-30 year periods is at the low edge for all ensemble members. The observed realization could be a rather rare event; this low observed PSD for 8-30 periods is within the spread of some ensembles, but I think another possibility is that the ensembles are all reacting too strongly to non-GHG forcing.
 - Fig S2: Similar comment to that from Fig 1b/c: For all but the MPI ensemble, the observed GMSST dip in 1991 is much smaller than the dip from almost all ensemble members.

I am also reminded of other evidence in Atlantic variability in the CESM LE that the model is responding too strongly to aerosol forcing, particularly from a paper by Kim et al. (2018): <https://journals.ametsoc.org/doi/pdf/10.1175/JCLI-D-17-0193.1>. Might be worth citing this study also at L244.

2. I realize that it is somewhat outside the scope of this study, but it might be useful to briefly cite studies in the introduction on how Atlantic variability modulates GMST variability, but on slightly longer timescales. This topic is briefly touched upon in the conclusions, but (in my opinion), could stand another sentence or two of discussion in the introduction since it is likely that forced Atlantic variability may also be overestimated in models.

3. I am a little bit concerned with the quadratic trend removal (though this is vastly better than the linear trend removal that I see in many other studies). How much of the multidecadal aerosol forcing is removed by the quadratic trend removal. Would this affect the power spectra estimates for the longer 20-30 year periods? For the decadal volcanic downturns examined here, I imagine that this is probably not a large concern.

Typos:

In the captions for Fig 2, S2, subfigures should be singular subfigure.

Caption 2: I think the word "integrating" in the last full sentence may be superfluous. Found this sentence confusing to get through, suggest revising.

Response to Reviewer's #1

Reviewer #1 (Remarks to the Author):

Review of the manuscript "A joint role for forced and internally-driven variability in the decadal modulation of global warming" by G. Liguori et al.
This paper presents the effects of volcanic aerosol forcing in the decadal variations of global mean surface temperature. By using large initial-condition ensembles with Earth System Models, they separate anthropogenic, natural, and internally-driven components of decadal variability. They found the volcanic aerosols account for ~29-53% of the decadal variance over 1950-2010, while externally forced decadal variance is very weak during 2010-2070 due to the lack of volcanic forcing. In addition, they also confirm the possible projection of external forcing onto the IPO variability. The paper is well-written and well-organized. However, some drawbacks should be addressed before the paper can be considered for publication in Nature Communications. Thus, the paper needs a major revision.

We sincerely thank the referee for reviewing our manuscript and providing many excellent and constructive suggestions for improving the overall quality of the manuscript. A detailed report describing how the comments were addressed can be found below. We use style and colour code as follows:

Italic/blue: for reviewer comment

Italic/black: manuscript text

Regular/black: Answer to the reviewer

Italic/magenta: Proposed change in the manuscript text

Major comments:

1. In this paper, both IPO and PDV are used to describe the decadal variability in the Pacific. Do you regard them as the same thing in this paper or not? Why do you use PDV in some places but IPO in others? Please clarify. For example, in Lines 71-76, the two sentences provide similar information, but the former uses IPO while the latter uses PDV. What is the difference between IPO and PDV?

We thank the reviewer for pointing this out. We agree that the lack of any explanation of what we mean by PDV and IPO generates confusion, especially in the text lines highlighted by the reviewer. We decided to drop the use of PDV except in the last paragraph, where we inform the reader that IPO is only one contributor (the largest) to the broader concept of Pacific decadal variability (we also direct the reader to the valuable work on the topic by Newman et al., 2016)

2. Besides the effect of volcanic forcing, the importance of internal variability can be inferred in the results of this paper. I suggest the authors adding some analysis and discussion to compare the contributions from internal component and volcanic forcing to the decadal modulation of GMSST quantitatively, not only for the variance as you have mentioned but also for the time evolution.

We appreciate the reviewer's suggestion and added a supplementary figure (Fig. S2) showing the correlation (as a boxplot) between GMSSTr ensemble mean and each

individual member for both historical and future period. In addition, the figure indicates the correlation value between each model ensemble mean and the observation. Thus, when this value falls within the range indicated by the boxplot, the model is assumed to be consistent with the observation. With this criterion, only two out of five models are formally consistent, with the majority of models that overestimate the externally-forced decadal variability and/or underestimate the range of observed internal variability. These conclusions are similar to the one presented at the end of the section 3 (“Role of aerosols, GHG, and biomass burning in decadal variability”).

As a result, the following text has been added to the section 2:

Neglecting any residual internal variability in the ensemble mean, the correlation between ensemble mean and each individual member gives a direct measure of the GMSSTr forced component (Fig. S2). Depending on the ensemble the forced component explains between 30 to 58% of the variance (explained variance obtained as correlation squared) in the historical period and 8 to 18% in the future period, with the remaining variance associated with unforced internal variability. However, three out of five models show correlations between observed and ensemble-mean time series outside the range of modelled internal variability (i.e., outside the range of correlations between the ensemble-mean and individual ensemble members). This may be a result of models underestimating the range of natural variability, but, as we show below, it is likely to also indicate an overestimation of the forced component in the historical period. (e.g., Fig. S4).

Fig. S2. Correlation between ensemble mean and individual members of the 8-year low-passed GMSST residual time series presented in Fig. 1b for historical (Hist) and future (Fut) periods. On each box-plot, the central mark indicates the median, the bottom and top edges of the box indicate the 25th and 75th percentiles, respectively, and the whiskers indicate minimum and maximum values. The black dot indicates the correlation between ensemble mean and observation.

3. By comparing between historical and future periods, the effect of volcanic forcing is discussed. Besides volcanic aerosols, are there any other forcing differences between the two periods that can cause differences in decadal variability? How about the anthropogenic aerosol forcing? Are there any decadal variations in anthropogenic aerosol forcing in historical or future periods? Please clarify.

We understand the reviewer’s concern that other external forcing rather than volcanic may cause differences between historical and future periods. While the CESM fixed-aerosol experiments presented in Fig. 3c seems to exclude an important role for the aerosol forcing

in the historical period, it does not preclude a significant role in future periods. However, the evolution of GMSSTr in this experiment until 2070 (being 2080 last year available) confirms that forced decadal fluctuations (i.e., coherent between all-forcing and fixed-aerosol experiments) only appear in the historical period, which is when the volcanic forcing is present (figure below).

Fig. 1_r1. Role of the aerosol forcing in CESM1. GMSSTr time series in (a) the “all forcing” ensemble (All-FORC, 20 members), and (b) the all forcing but fixed anthropogenic aerosols (Fixed-AERO, 20 members). In each subfigures grey lines indicate ensemble members, the black line indicates the ensemble mean, and the green line indicates the ensemble mean in the All-FORC. Units are in [°C].

In addition, while the anthropogenic aerosol forcing differs between historical and future periods, its evolution in the 21st century presents a rather smooth negative trend that is difficult to link with the kind of decadal-scale fluctuations analysed in this study. Here is a figure from the chapter 8 of synthesis report of the IPCC Fifth Assessment Report (AR) showing the aerosol forcing in CMIP5 experiments for the RCP8.5 scenario, which is also the one used in LENS simulations.

Figure 8.22 | Global mean anthropogenic forcing with symbols indicating the times at which ACCMIP simulations were performed (solid lines with circles are net; long dashes with squares are ozone; short dashes with diamonds are aerosol; dash-dot are WMGHG; colours indicate the RCPs with red for RCP8.5, orange RCP6.0, light blue RCP4.5, and dark blue RCP2.6). RCPs 2.6, 4.5 and 6.0 net forcings at 2100 are approximate values using aerosol ERF projected for RCP8.5 (modified from Shindell et al., 2013c). Some individual components are omitted for some RCPs for visual clarity.

Moreover, we acknowledge that other studies have suggested an important role for tropospheric aerosol in both global mean surface temperature and IPO variability. Specifically, in the opening paragraph we present this possibility and refer the reader to the seminal study of Smith et al., (2016), which suggests the possibility that the early 2000s hiatus in the GMST increase was driven by changes in atmospheric aerosols. While our study does not focus on the warming hiatus, it must be noted that our five large ensembles do not show any consistent slowdown in the temperature during the hiatus period (i.e., 1998–2013), suggesting for this event a dominant role of the internal variability over the externally-forced variability.

4. Lines 126-128: The decadal variations seem not very similar between observation and ensemble means. Why are the decadal variations similar among models, but different from the observation? They should have the same externally forced variations. Does this difference indicate the effect of the internal component in the observed variability? The correlation coefficients between observation and ensemble means can indicate the percent of decadal variations explained by volcanic forcing.

We agree with the reviewer that lines 126-128 do not accurately describe the differences between observation and ensemble mean. We have changed that statement that now reads:

“The observed GMSSTr trajectory lies largely within the ESM-LE model spread envelopes and presents some similarities with GMSSTr ensemble means from about 1975 to 1995, with the lack of full agreement largely due to the internal variability but also to model deficiencies in representing the external forcing (discussed later).”

The lack of similarity between modelled and observed GMSSTr trajectory is largely due to the internal variability, but also to model deficiencies in representing the external forcing. As noted by the reviewer in their point#5, and clearly visible in the previous Fig. S2, now Fig. S4, models tend to overestimate the cooling response after major volcanic events, causing the simulated trajectories to diverge from the observed one. It is noteworthy that while the amplitude of the decadal fluctuations is partially off, the timing is often correct (see the drop in the observed time series after the three major events in the study period, Fig. 1b and c).

While it is impossible to fully quantify the role of the volcanic forcing without a dedicated ensemble (as also suggested by the review’s point# 8), we believe that our calculations for the percent of externally-forced decadal variations obtained integrating the power spectrum of Fig. 3, provides an upper bound for the amount of variance in GMSSTr driven by volcanic eruption, acknowledging that other minor contributions may come from other external forcings (e.g., tropospheric aerosols).

We thank the reviewer for their advice on looking at the correlation between observation and ensemble means. Their suggestion was well taken and led to an in-text discussion and a supplementary figure (see our answer to reviewer’s comment #2). We show the correlation (as a boxplot) between GMSSTr ensemble mean and each individual member, together with the correlation value between each model ensemble mean and the observation. We find that only two out of five models are consistent with the observations, as the majority of models either overestimate the externally-forced decadal variability and/or underestimate the range of observed internal variability

5. Lines 162-165: Similar to the last comment, the decadal variations from volcanic eruptions are obvious only in the ensemble means but not in observation, again indicating the importance of internal variability in observation. This should be clarified in the text.

We thank the reviewer for their well-pointed comment. We now clarify multiple points (see our answers to reviewer’s comment #2, and #4, and text added section 2 copied below) that the internal variability seems to play a more important role in the observation than in the models, as consequence of the model tendency for overestimating the volcanic response (Fig. 3S), which may be simply the result of inaccuracy in the external forcing (Kravitz & Robock, 2011; Santer et al., 2014; Toohey et al., 2011).

Kravitz, B., & Robock, A. (2011). Climate effects of high-latitude volcanic eruptions: Role of the time of year. *Journal of Geophysical Research: Atmospheres*, 116(D1). <https://doi.org/10.1029/2010JD014448>

Santer, B. D., Bonfils, C., Painter, J. F., Zelinka, M. D., Mears, C., Solomon, S., et al. (2014). Volcanic contribution to decadal changes in tropospheric temperature. *Nature Geoscience*. <https://doi.org/10.1038/ngeo2098>

Toohey, M., Krüger, K., Niemeier, U., & Timmreck, C. (2011). The influence of eruption season on the global aerosol evolution and radiative impact of tropical volcanic eruptions. *Atmospheric Chemistry and Physics*, 11(23), 12351–12367. <https://doi.org/10.5194/acp-11-12351-2011>

However, it must be noted that the observed power spectrum (black lines in Fig. 2) for periods between 8 and 30 years is lower than almost all ensemble members (grey lines in Fig. 2), independently of the model. While it possible that the observed GMSSTr trajectory is a rare event, the discrepancy is more likely a symptom of a systematic bias in decadal variability, probably linked to the model tendency to overestimate the non-GHG forced component (e.g., Fig. S2 and Fig. S4).

6. Lines 180-183: This statement is interesting and needs to be clarified. Why aerosols from troposphere have offset effect to those from stratosphere? Do you have any conjecture or related references to support this statement?

Lines 180-183: *It is noteworthy that fixing tropospheric anthropogenic aerosol concentrations to their 1920 value results in larger amplitude fluctuations in GMSSTr, suggesting that higher tropospheric aerosol concentrations typical of the second half of the 20th Century, have a damping effect on the volcanic forcing.*

This is a very good point that deserves a full study. At the moment we do not have an explanation for this result except the following possible line of reasoning: in the limit of a troposphere saturated with aerosols, any further addition in the stratosphere (e.g., volcanic aerosols) would not be able to affect the Earth's surface, as the incoming solar radiation would be already screened by the saturated troposphere.

Given the importance of this result, we plan a dedicated study, the first step of which will be to verify that similar effects can be seen in other models.

7. Lines 216-217: *“Assuming the IPO to be purely a result of internal variability, one would expect this IPO variance to decrease in proportion to the number of ensemble members, n .” How did you get this expectation? I cannot understand. Please explain.*

We agree that an explanation for this expectation/result is needed. In the new version of the manuscript we added the following paragraph in the section **Methods**

A null hypothesis for the IPO variance in the ensemble mean

Assuming the IPO to be purely a result of internal variability, thus independent in each ensemble member, one can use a known statistical result to predict the ensemble mean IPO variance as function of the number of ensemble members.

Consider N independent time series denoted X_i , $i = 1, 2, \dots, N$, each with variance $Var(X_i) = \sigma^2$, $i = 1, 2, \dots, N$. Now the variance for the average of the N time series is $Var\left(\frac{1}{N}\sum_{i=1}^N X_i\right) = \frac{1}{N^2}\{\sum_{i=1}^N Var(X_i) + \sum_{i \neq j}^N Cov(X_i, X_j)\}$, where $Cov(X_i, X_j)$ is the covariance between the time series X_i and X_j , $i, j = 1, 2, \dots, N$. Since the time series are assumed to be independent, this last term is equal zero and the variance for the average reduces to $Var\left(\frac{1}{N}\sum_{i=1}^N X_i\right) = \frac{1}{N^2}N\sigma^2 = \frac{\sigma^2}{N}$, namely the variance for each independent timeseries, σ^2 , scaled by N .

8. Fig. 3: Why don't you provide the results for simulations with fixed volcanic forcing here? That will be more straightforward to prove its effect in dominating the decadal variations in GMSST.

We fully agree with the reviewer that fixed volcanic forcing simulations would be ideal, however, a large ensemble for such an experiment is unavailable. Running such an ensemble is computationally expensive, requires significant storage capability, and it is outside the scope of the study. While a large ensemble is unavailable, several years ago NCAR's scientists used an early version of CESM to produce few simulations in which the volcanic forcing was either the only one present (i.e., volcanic-forcing-only experiment; 5 members) or the only one excluded (i.e., all-but-volcanic-forcing experiment; 4 members). While the small number of members does not allow to effectively extract the forced variability, these experiments seem to be consistent with the hypothesized key role for volcanic forcing, as shown by the GMSSTr time series presented below and added to the supplementary material (Fig. S3).

This new text has been added to the manuscript:

A direct estimate of volcanic-driven GMSST variability requires large ensembles of simulations in which the volcanic forcing is either the only one present (i.e., volcanic-forcing-only experiment) or the only one excluded (i.e., all-but-volcanic-forcing experiment). While such experiments are unavailable in CESM-LE, results from smaller ensembles (4 and 5 members) with a similar version of CESM are consistent with the hypothesized role for volcanic forcing (Fig. S3).

Fig. S3. Role of the volcanic forcing in CESM1. GMSSTr time series in (a) the “all forcing” ensemble (All-FORC, 20 members), (b) the “all-but-volcanic-forcing” ensemble (All-but-VOLC, 4 members), and (c) the “volcanic-forcing-only” ensemble (VOLC-only, 5 members). In each subfigure grey lines indicate ensemble members, the black line indicates the ensemble mean, and the green line indicates the ensemble mean in the All-FORC. Units are in [°C].

Minor comments:

1 Lines 216 and 220, Fig. S2 should be Fig. 4c.

We thank the reviewer for noticing this.

2 Caption of Fig. 3: The second “b” should be “d”

We thank the reviewer for noticing this.

Response to Reviewer's #2

Reviewer #2 (Remarks to the Author):

Review of "A joint role for forced and internally-driven variability in the decadal modulation of global warming" by Liguori et al.

This manuscript examines how much decadal-timescale variability in Earth system model Large Ensembles is due to external forcing and how that amount compares to the decadal variability in observations. By using Large Ensembles in both the historical and future period, the authors are able to assess how much decadal-timescale variability is externally forced (using the ensemble mean) versus internally generated (by examining all ensemble members). They trace much of the decadal variability in modelled global warming during the historic period to volcanic aerosols. The addition of results from the CESM all-but-one-forcing ensembles further illustrate this possibility in a convincing way. In these Large Ensembles, volcanic eruptions tend to excite the IPO pattern, indicating that much variability in the IPO may be externally forced. From there, the manuscript concludes that modelled decadal variability in the future may be unpredictable given the unpredictable nature of volcanic Eruptions.

I very much appreciate the approach of using multiple Large Ensembles. For studies examining climate variability, this approach is still quite novel (we've only had convenient access to multiple Large Ensembles for the last year) and should be a gold-standard method going forward for examining these kinds of questions in climate models. This method gets at both models' representation of internal variability and allows an assessment of structural model differences, as the authors have done here. Because of the rigor of using Large Ensembles along with strong statistical methods, I find the conclusions drawn from this study convincing and of relevance to the broader climate community. My one suggestion to the authors is to further flesh out another possible conclusion that is implied at multiple points in the manuscript: that climate models as a whole are reacting too strongly to aerosols and that forced IPO variability is too strong overall as compared to observations (see discussion below). In the figures, I see some support for this argument and think that the authors would only strengthen their manuscript by pointing out this possibility more explicitly.

The figures are of publication quality and the writing is very clear with almost no typos. I recommend acceptance after addressing a few (very) minor concerns and suggestions, listed below.

We sincerely thank the referee for reviewing our manuscript and providing many excellent and constructive suggestions for improving the overall quality of the manuscript. A detailed report describing how the comments were addressed can be found below. We use style and color code as follows:

Italic/blue: for reviewer comment

Italic/black: manuscript text

Regular/black: Answer to the reviewer

Italic/magenta: Proposed change in the manuscript text

Minor concerns/suggestions:

1. As discussed above, I think that the results of this manuscript suggest that all of these ensembles have too strong responses to volcanic aerosol forcing (besides just CANESM2 and GFDL) and that there is a structural model deficiency across all the ensembles in the

same direction. I suggest to the authors to address this possibility a bit more in the conclusions section. Here's the evidence I see in the figures for this possibility:
 - Fig 1b/c, 1991. The observed dip in residual GMSST is at the very upper edge or outside the spread of all ensembles. While it is possible that 1991 was a rare event, the fact that observations lie outside the spread of all 5 ensembles (in 1b) suggests to me that all 5 ensembles have something in common that is leading them to have a similar (possibly incorrect) response.

Following suggestions from both reviewers, we now addressed more explicitly the possibility that models overestimate the response to the external forcing. We added an in-text discussion, a new figure to the supplementary materials (Fig. S2), and an additional statement in the conclusions section.

From section 2:

The observed GMSSTr trajectory lies largely within the ESM-LE model spread envelopes and presents some similarities with GMSSTr ensemble means from about 1975 to 1995, with the lack of full agreement largely due to the internal variability but also to model deficiencies in representing the external forcing (discussed later).

...

Neglecting any residual internal variability in the ensemble mean, the correlation between ensemble mean and each individual member gives a direct measure of the GMSSTr forced component (Fig. S2). Depending on the ensemble the forced component explains between 30 to 58% of the variance (explained variance obtained as correlation squared) in the historical period and 8 to 18% in the future period, with the remaining variance associated with unforced internal variability. However, three out of five models show correlations between observed and ensemble-mean time series outside the range of modelled internal variability (i.e., outside the range of correlations between the ensemble-mean and individual ensemble members). This may be a result of models underestimating the range of natural variability, but, as we show below, it is likely to also indicate an overestimation of the forced component in the historical period. (e.g., Fig. S4).

Fig. S2. Correlation between ensemble mean and individual members of the 8-year low-passed GMSST residual time series presented in Fig. 1b for historical (Hist) and future (Fut) periods. On each box-plot, the central mark indicates the median, the bottom and top edges of the box indicate the 25th and 75th percentiles, respectively, and the whiskers indicate minimum and maximum values. The black dot indicates the correlation between ensemble mean and observation.

From section 5 (Conclusions):

While the forced variability is visible in the observations, it must be noted that models present too much decadal variability (Fig. 2) and likely overestimate the response to the external forcing (Fig. S2 and S4).

In addition, we think that some of the reviewer's suggestions are captured in these statements taken from the section 3 ("Role of aerosols, GHG, and biomass burning in decadal variability"):

Comparing GMSST anomalies after each major volcanic eruption suggests a tendency for a colder-than-observed response in ESM-LE models during the 1982 and 1991 events (Fig. S3)... Furthermore, models that likely overestimate the volcanic response, CANESM2 and GFDL, present also the highest percentage of decadal variability accounted for by FDV (53%; Fig. 2a,l), suggesting that these models might overestimate the externally-forced fraction of decadal variability, which is therefore closer to the lower boundary of the range estimated in ESM-LE (i.e., [29-53%]).

- Fig 2, left column: The observed power spectral density (black lines) is lower than the ensemble mean (forced) variability and lower than almost all the ensemble members (gray lines) in all 5 ensembles for periods ≥ 8 years. I find it rather unlikely that across all 5 ensembles with 150+ possible realizations, that the observed realization at 8-30 year periods is at the low edge for all ensemble members. The observed realization could be a rather rare event; this low observed PSD for 8-30 periods is within the spread of some ensembles, but I think another possibility is that the ensembles are all reacting too strongly to non-GHG forcing.

We find the reviewer's comment very valuable and incorporated its essence in the following new lines of section 2:

However, it must be noted that the observed power spectrum (black lines in Fig. 2) for periods between 8 and 30 years is lower than almost all ensemble members (grey lines in Fig. 2), independently of the model. While it is possible that the observed GMSSTr trajectory is a rare event even with these multiple large ensembles, the discrepancy is more likely a symptom of the model tendency to overestimate internal and/or non-GHG forced decadal variability (e.g., Fig. S2 and Fig S4).

- Fig S2: Similar comment to that from Fig 1b/c: For all but the MPI ensemble, the observed GMSST dip in 1991 is much smaller than the dip from almost all ensemble members.

While we agree with the reviewer's comment, we think that the following text from the section 3 captures the essence of the comment:

Comparing GMSST anomalies after each major volcanic eruption suggests a tendency for a colder-than-observed response in ESM-LE models during the 1982 and 1991 events (Fig. S3).

...

Here with 30 members for each model we see that in most ESM-LE simulations, and in all members from CANESM2 and GFDL ensembles, the magnitude of the global temperature response to the 1982 and 1991 volcanic events is overestimated regardless of the El Niño–Southern Oscillation (ENSO) phase (Fig. S4). Furthermore, models that likely overestimate the volcanic response, CANESM2 and GFDL, present also the highest percentage of decadal variability accounted for by FDV (53%; Fig. 2a,l), suggesting that these models

might overestimate the externally-forced fraction of decadal variability, which is therefore closer to the lower boundary of the range estimated in ESM-LE (i.e., [29-53%]).

I am also reminded of other evidence in Atlantic variability in the CESM LE that the model is responding too strongly to aerosol forcing, particularly from a paper by Kim et al. (2018): <https://journals.ametsoc.org/doi/pdf/10.1175/JCLI-D-17-0193.1>. Might be worth citing this study also at L244.

We thank the reviewer for the suggestion. The reference has been included.

2. I realize that it is somewhat outside the scope of this study, but it might be useful to briefly cite studies in the introduction on how Atlantic variability modulates GMST variability, but on slightly longer timescales. This topic is briefly touched upon in the conclusions, but (in my opinion), could stand another sentence or two of discussion in the introduction since it is likely that forced Atlantic variability may also be overestimated in models.

The reviewer's point is a readily shareable opinion. In fact, in an early draft of the manuscript we briefly discussed the role of the Atlantic multidecadal variability in GMST. However, given the format of the journal, we preferred a shorter and more concise introductory section. Moreover, while the Atlantic variability is not explicitly named in the introduction section, it is implicitly included in some of the references (Meehl et al., 2013; Haustein et al., 2019).

3. I am a little bit concerned with the quadratic trend removal (though this is vastly better than the linear trend removal that I see in many other studies). How much of the multidecadal aerosol forcing is removed by the quadratic trend removal? Would this affect the power spectra estimates for the longer 20-30 year periods? For the decadal volcanic downturns examined here, I imagine that this is probably not a large concern.

We fully understand the reviewer's concern as this was also of our concern in early stages of the study. However, except for possible effects at the edges of the study period (i.e., 1950-2070), the removal of a smooth centennial-scale signal fitted by the quadratic function has a little impact on the decadal-scale fluctuations targeted in this study. The large difference in the timescales involved, centennial vs decadal, prevent the creation of spurious decadal-scale fluctuations far from the edges of the study period. Moreover, while the quadratic trend removal clearly affects the shape of the power spectrum (the quadratic fit has power at all frequencies), it does not significantly affect our findings on the forced decadal variability.

We now note possible effects of the trend removal with these in lines of section 2:

Moreover, the removal of a centennial-scale quadratic trend does not significantly affect the shorter decadal-scale fluctuations targeted in this study, apart from possible effects at the edges of the study period (i.e., 1950-2070).

Typos:

In the captions for Fig 2, S2, subfigures should be singular subfigure.

We thank the reviewer for noticing this.

Caption 2: I think the word "integrating" in the last full sentence may be superfluous. Found this sentence confusing to get through, suggest revising.

We agree with the reviewer suggestion and acted accordingly by rephrasing the sentence

REVIEWERS' COMMENTS second round:

Reviewer #1 (Remarks to the Author):

Review of the manuscript "A joint role for forced and internally-driven variability in the decadal modulation of global warming" by G. Liguori et al.

The authors have satisfactorily responded to most of my initial questions and comments. I only have several comments to be answered before publication. The specific comments are shown below.

Specific comments:

1. Lines 126-129: Do you have references for the region choice?
2. Lines 198-199: Why does it need so long time to recover? Do you have any explanation or reference?

Reviewer #2 (Remarks to the Author):

The authors have addressed my and the other's reviewers concerns and suggestions. The revisions have strengthened the manuscript. I recommend its acceptance, and that the following edits be made somewhere in the proofing stage:

L133: remove "from"

L231: add tilde to Niño.

L260: at *a* lower rate

L286-288: This statement is controversial. The true breakdown between forced/internally-varying AMV in observations is difficult to quantify. It is more defensible to state that aerosols are driving most of the *simulated* AMV.

Response to Reviewer's #1 (Round#2)

Reviewer #1 (Remarks to the Author):

*Review of the manuscript "A joint role for forced and internally-driven variability in the decadal modulation of global warming" by G. Liguori et al.
The authors have satisfactorily responded to most of my initial questions and comments. I only have several comments to be answered before publication. The specific comments are shown below.*

We thank again the referee for reviewing our revised manuscript and providing additional valuable comments. We use style and colour code as follows:

Italic/blue: for reviewer comment

Italic/black: manuscript text

Regular/black: Answer to the reviewer

Italic/magenta: Proposed change in the manuscript text

Specific comments:

1. Lines 126-129: Do you have references for the region choice?

The reference for the region has been added.

Smith, T. M., and R. W. Reynolds, 1998: A high resolution global sea surface temperature climatology for the 1961–90 base period. *J. Climate*, **11**, 3320–3323.

2. Lines 198-199: Why does it need so long time to recover? Do you have any explanation or reference?

The temporal spacing between these strong drops in temperature, 19 and 9 years, combined with a recovery time (i.e., time to dissipate the cold anomaly) of 5-8 years (Fig. 1c), creates a climate signal with a strong projection on decadal-scale variability.

Large volcanic eruptions eject sulfur particles into the stratosphere that are rapidly converted to sulfate aerosols. The aerosol act diminishing the net incoming solar flux at the top of the atmosphere, resulting in a cooling of the Earth surface.

While these sulfate aerosols have a residence time of about 1–2 years in the stratosphere (Robock 2000) they can cause surface cooling for many more years after the eruption. Volcanically induced cooling of the ocean surface penetrates into deeper layers where it persists for years after the event. Gupta and Marshall (2018), estimate that most of the surface temperature anomaly is dissipated within 10 years (see also the Fig. 1 below taken from Gupta and Marshall, 2018). We now cite the work of Gupta and Marshall (2018) in association to the recovery timescale visible in Fig. 1c of the manuscript.

FIG. 1. Surface temperature response of the box model to an idealized Pinatubo eruption (-4 W m^{-2} for a year) in the 1- (red) and 2-box cases (blue) in terms of the ratio of ocean mixing strength to the climatic feedback parameter $\mu = q/\lambda$ with $\lambda = 1.5 \text{ W m}^{-2} \text{ K}^{-1}$. All other parameters are as in Table 1. The “area under the curve” is the same in all cases when integrating to infinity, but with a smaller peak and a longer “tail” as q (or μ) increases.

Robock, A., 2000: Volcanic eruptions and climate. *Rev. Geophys.*, 38, 191–219, <https://doi.org/10.1029/1998RG000054>.

Gupta, M. and J. Marshall, 2018: The Climate Response to Multiple Volcanic Eruptions Mediated by Ocean Heat Uptake: Damping Processes and Accumulation Potential. *J. Climate*, 31, 8669–8687, <https://doi.org/10.1175/JCLI-D-17-0703.1>

Response to Reviewer's #2 (Round#2)

Reviewer #2 (Remarks to the Author):

The authors have addressed my and the other's reviewers concerns and suggestions. The revisions have strengthened the manuscript. I recommend its acceptance, and that the following edits be made somewhere in the proofing stage:

We thank again the referee for reviewing our revised manuscript and providing additional valuable comments. We agree to all these changes and have modified the text accordingly